# The Mechanism of Ubiquitination or Deubiquitination Modifications in Regulating Solid Tumor Radiosensitivity

**DOI:** 10.3390/biomedicines11123240

**Published:** 2023-12-07

**Authors:** Mengyun Zhang, Yingjie Shao, Wendong Gu

**Affiliations:** Department of Radiation Oncology, The Third Affiliated Hospital of Soochow University, Changzhou 213003, China; 20225235102@stu.suda.edu.cn

**Keywords:** ubiquitination, deubiquitination, radiosensitivity, cancer

## Abstract

Radiotherapy, a treatment method employing radiation to eradicate tumor cells and subsequently reduce or eliminate tumor masses, is widely applied in the management of numerous patients with tumors. However, its therapeutic effectiveness is somewhat constrained by various drug-resistant factors. Recent studies have highlighted the ubiquitination/deubiquitination system, a reversible molecular modification pathway, for its dual role in influencing tumor behaviors. It can either promote or inhibit tumor progression, impacting tumor proliferation, migration, invasion, and associated therapeutic resistance. Consequently, delving into the potential mechanisms through which ubiquitination and deubiquitination systems modulate the response to radiotherapy in malignant tumors holds paramount significance in augmenting its efficacy. In this paper, we comprehensively examine the strides made in research and the pertinent mechanisms of ubiquitination and deubiquitination systems in governing radiotherapy resistance in tumors. This underscores the potential for developing diverse radiosensitizers targeting distinct mechanisms, with the aim of enhancing the effectiveness of radiotherapy.

## 1. Introduction

Tumors not only have a low cure rate and poor prognosis but are also associated with high morbidity and mortality rates globally, becoming one of the leading causes of death worldwide [1]. Among the various treatment modalities for tumors that are emerging, radiotherapy is promising with the help of real-time imaging, which can achieve precise localization; moreover, the use of radioactive rays can induce single- or double-strand breaks (DSBs) in the cellular DNA of target tumors [2,3]. This approach kills tumor cells or inhibits their growth in the most effective manner and causes minimal damage to normal tissues, which benefits many patients with tumors. However, in clinical workup, not all patients with tumors are sensitive to radiotherapy or respond well to this treatment, often because of tumor heterogeneity [4] and dynamic changes in the tumor microenvironment [5,6,7]. This can lead to radiation tolerance in some patients after a period of standard radiotherapy and, subsequently, tumor recurrence and/or distant metastasis. The regulatory network of tumor cell radiosensitivity is intricate and complex, and the related cross-links play vital roles, such as in DNA damage repair, cell cycle, cancer stem cells, cell signaling pathway disorders or inactivation, the hypoxic tumor microenvironment, and related metabolic disorders [8,9,10,11]. Therefore, exploring key molecules in the network of radiosensitivity regulatory mechanisms, assessing effective and low-toxicity radiosensitizers, identifying biomarkers that can predict the efficacy of radiotherapy, realizing individualized treatment, and improving the local control rate of malignant tumors have always been research directions to promote the development of radiotherapy.

In human cells, molecular modifications, such as m6A modification [12] and phosphorylation [13], dynamically regulate molecular function and stability in time and space by “remodeling” the molecular structure [14,15]. Recently, other modifications such as ubiquitination and deubiquitination have attracted the attention of researchers. Ubiquitination is a dynamic, multi-step enzymatic cascade reaction that tags proteins with ubiquitin; in this process, ubiquitin is first activated by an E1 enzyme (ubiquitin-activating enzyme), after which the E1 enzyme passes the activated ubiquitin to an E2 enzyme (ubiquitin-conjugating enzyme), and the activated ubiquitin is then ligated or tagged to a substrate protein catalyzed by an E3 enzyme (ubiquitin ligase) [16] which, in turn, catalyzes downstream biochemical reactions. Various ubiquitin-catalyzing enzyme systems possessing different structures have been identified, including a wide variety of E3 ligases, indicating the molecular specificity of the ubiquitination pathway [17] (Figure 1). These systems are crucial for the precise regulation of cell growth, development, and stability. In most previous studies, it was demonstrated that ubiquitin molecules primarily exert their function by binding with Lys residues, leading to different outcomes depending on the modified site. For instance, the degradation of substrate proteins through the proteasome predominantly hinges on K48-linked ubiquitin chains [18]. In contrast, K63 ubiquitin chains play a pivotal role in mediating biological functions, such as DNA damage repair [19,20,21]. Furthermore, recent research has confirmed that aside from Lys, modifications targeting Ser side chains can also occur indirectly [22].

When researchers initially discovered ubiquitin [23], it was widely believed that ubiquitination modifications primarily mediated protein degradation, regulating the levels of cellular molecules [24]. However, through comprehensive studies on ubiquitination’s function, it has been discovered that ubiquitin also possesses non-proteolytic functions. It has been established that it plays widespread roles in various biological processes, including its involvement in cell signaling pathways, DNA damage repair, cell division, and endocytosis [25,26,27]. These biological processes are also pivotal in the development of tumor cells. This significant finding suggests that ubiquitination modifications may play crucial roles in the proliferation, migration, invasion, and even the therapeutic resistance of tumor tissues, potentially serving as biomarkers for tumor diagnosis, therapeutic targets, and prognosis prediction [28,29]. Notably, E3 ligase, a key enzyme in the final step, holds promise as a drug target for tumor treatment, and investigations into potential anticancer drugs targeting relevant components of the ubiquitination system, including E3 ligases, are underway [30,31].

Human cells also harbor a deubiquitination system catalyzed by the enzyme deubiquitinase (DUB), which removes ubiquitin from substrate proteins, thereby reversing substrate ubiquitination [32]. The ubiquitination and deubiquitination systems work dynamically to regulate protein stability and collectively maintain an organism’s normal biological processes. Similarly, if the key enzymes of this dynamic system are genetically mutated or dysfunctional, they can induce varying degrees of pathophysiological disturbances in the organism, ultimately leading to the development of diseases [32].

This review centers on several prevalent tumors primarily treated with radiotherapy. It delves into the effects of ubiquitination or deubiquitination modifications on specific proteins and their influence on the radiosensitivity of these tumors (Table 1) (Figure 2). Additionally, it outlines the clinical potential of targeting ubiquitinated or deubiquitinated protein pathways to enhance the radiosensitivity of tumor cells.

## 2. Mechanisms by Which Ubiquitination and Deubiquitination Systems Affect Tumor Radiosensitivity

### 2.1. Nasopharyngeal Carcinoma

In the context of nasopharyngeal carcinoma (NPC), a prevalent malignancy among head and neck tumors, radiotherapy stands as a cornerstone treatment [91]. Pertinent studies have elucidated the pivotal roles played by ubiquitination enzymes and DUBs in modulating NPC radiosensitization. Researchers have observed an elevated expression of the E3 ligase TRIM21 in NPC which propels tumor cell proliferation. Additionally, it was discovered that, aided by SERPINB5, radiation intensifies the TRIM21-mediated ubiquitin degradation of GMPS. This culminates in the suppression of downstream TP53 expression, subsequently conferring radioresistance to tumor cells [33]. Another investigation unearthed an alternate mechanism through which TRIM21 hinders radiation-induced antitumor immune responses. This occurs through its mediation of VDAC2 ubiquitination degradation, leading to the inhibition of mitochondrial DNA release [34]. Similarly, the E3 ligase RNF8 was shown to play a crucial role at DNA damage sites in NPC cells. By recruiting and ubiquitinating various factors such as Chk1 and Chk2, RNF8 facilitates DNA damage repair, consequently promoting the resistance of NPC cells to radiotherapy [35]. Another study pointed out the interconnection between ubiquitination and cellular metabolism in determining radiosensitivity. Specifically, HILPDA was found to modulate mitochondrial cardiolipin levels by impeding the PINK1-mediated ubiquitin degradation of CLS1. This process, in turn, bolstered mitochondrial autophagy, thereby heightening the radioresistance of NPC [92]. Additionally, FBP1 was identified as an inhibitor of the auto-ubiquitination of the E3 ligase FBXW7, inducing FBXW7 to inhibit the mTOR pathway. This dual action repressed glycolysis and amplified the radiosensitivity of NPC [36]. Chen et al. discovered that the DUB USP44 exhibited a low expression in NPC cells; however, USP44 induced the radiosensitization of NPC cells both in vitro and in vivo. Further investigations have revealed that USP44 targets the E3 ligase TRIM25 for deubiquitination, leading to the downstream degradation of Ku80. This, in turn, hampers the DNA damage repair associated with Ku80. Simultaneously, USP44 regulates the cell cycle, and the synergistic effect of these actions induces apoptosis and heightens the radiosensitivity of NPCs [37]. Although it is widely accepted that radiation primarily induces apoptosis in tumor cells [93], researchers have also noted the morphological traits of pyroptosis in radiation-treated NPC cells. This process was subsequently found to be mediated by GSDME and induced by radiation through the intrinsic mitochondrial apoptotic pathway. Moreover, GSDME protein levels were found to be low in radiation-insensitive cancer tissues. Researchers conducted a series of experiments in NPC cells, demonstrating that the DUB OTUD4 stabilizes GSDME and heightens the radiosensitivity of NPC by promoting GSDME-dependent pyroptosis. This research suggested a novel clinical avenue of targeting the OTUD4/GSDME axis to induce pyroptosis, thereby enhancing NPC’s sensitivity to radiotherapy [38].

### 2.2. Esophageal Cancer

Esophageal cancer, especially the predominant subtype of esophageal squamous cell carcinoma (ESCC), is one of the most prevalent and lethal malignant tumors globally [1,94]. In the realm of ESCC treatment, particularly for locally advanced cases, radiotherapy emerges as a highly effective tool [95]. Multiple studies have substantiated the effect of ubiquitination-related enzymes on radiotherapy outcomes in esophageal cancer tissues, operating through diverse pathways. For instance, the ubiquitin-conjugating E2 enzyme variant UBE2D3 has been identified as a regulator of esophageal cancer radiosensitivity, influencing it through various mechanisms. Notably, UBE2D3 knockdown augments the expression and activity of telomerase hTERT, enhancing telomere stability. Simultaneously, it affects cell cycle control and DNA damage repair and induces radioresistance [39]. Another component, SOCS6, belonging to the E3 ubiquitin ligases, bolsters the radiosensitivity of ESCC tumors. Further investigations have revealed that SOCS6 influences tumor cell stemness primarily by targeting the ubiquitination degradation of c-Kit, leading to radiosensitization [40]. Previous studies have implicated the E3 ligase RAD18 in promoting migration and invasion in ESCC [96]. Subsequent research uncovered RAD18’s role in modulating ESCC radiosensitivity. In this case, RAD18, rather than acting as a ubiquitinating enzyme, enhances non-homologous end-joining (NHEJ)-mediated DSB repair rather than homologous recombination (HR), favoring radiotherapy resistance by upregulating DNA-PKc phosphorylation levels [41]. Similarly, SKP2, a substrate recognition subunit of the SCF^SKP2^ ubiquitin ligase complex, boosts DNA damage repair in cancer cells by regulating the expression of Rad51, a critical protein associated with the repair of DSBs. This leads to cellular radioresistance; however, the precise mechanisms governing Rad51 expression require further elucidation [97]. In a study by Dai et al., an examination of cancer tissues from 331 patients with ESCC compared with matched cancer-adjacent normal tissue samples revealed a significant reduction in the expression of the E3 ligase PELI1 in ESCC tissues. PELI1 was found to promote ionizing radiation (IR)-induced apoptosis. Further probing into the molecular mechanism revealed that PELI1 curtails NF-κB-inducing kinase (NIK) protein levels through ubiquitination, inhibiting the activation of the atypical NF-κB signaling pathway. This promotion of cancer cell apoptosis heightens tumor sensitivity to radiotherapy [42]. Suo et al. demonstrated that NRIP3 resisted radiation-induced cellular damage by upregulating the expression of DDI1 and fostering the formation of a complex combining DDI1 and RTF2. This, in turn, prompted the ubiquitination degradation of RTF2, allowing cells to restart their replication forks [98]. Similarly, another study showcased TRIB3’s role in promoting the cancer stem cell (CSC)-like properties of cancer cells by inhibiting the β-TrCP-mediated ubiquitination degradation of TAZ (a key downstream molecule of the Hippo pathway). This leads to induced radioresistance [43]. Additionally, in ESCC, SNPH exacerbates radiation-induced oxidative damage by mediating mitochondrial aggregation and redistribution. However, further examination of radioresistant ESCC cells revealed that SNPH can be ubiquitinated and degraded, contributing to cellular resistance to radiation. This study posited that weakly expressed SNPH might serve as a potential molecular indicator for predicting radiotherapy resistance, suggesting strategies targeting SNPH to enhance the efficacy of radiotherapy in ESCC [99]. As mentioned above, ionizing radiation induces the disruption of redox homeostasis through the generation of reactive oxygen species (ROS), which is one of the main mechanisms underlying the radiation-induced killing of tumor cells [100]. As confirmed in previous reports in the literature, the LKB1-AMPK axis plays a crucial role in regulating cellular metabolism, particularly in maintaining redox homeostasis [101]. So, can the LKB1-AMPK axis regulate the radiosensitivity of tumor cells? Researchers first found that the expression of the LKB1 protein was significantly increased in irradiated esophageal cancer cells and that LKB1 could induce the resistance of xenograft tumors in nude mice to radiation. Further mechanistic studies revealed that LKB1 primarily inhibits cell apoptosis and activates autophagy; both pathways together induce radioresistance in tumor cells, and this effect requires the involvement of AMPK [102]. However, another study identified an upstream regulatory molecule of the LKB1-AMPK axis: the E3 ligase RNF146. RNF146 mediates LKB1 ubiquitination to disrupt the formation of protein complexes between LKB1 and other proteins, thereby inhibiting LKB1 activation, rather than exerting proteasomal degradation functions [103]. Therefore, it is reasonable to speculate that the nondegradative ubiquitination of LKB1 by RNF146 may affect the regulation of esophageal cancer radiosensitivity by LKB1. However, further experimental validation is required to confirm this. Within the deubiquitination system, the DUB USP28 emerges as a regulator of cancer cell radiosensitivity through the c-Myc/HIF-1α axis [104].

### 2.3. Lung Cancer

Globally, lung cancer remains a highly prevalent and lethal malignancy [1]. Recent studies have revealed a diverse array of enzymes involved in ubiquitination, showing aberrant expression in lung cancer development. They play pivotal roles in regulating cell proliferation, metastasis, and apoptosis through different pathways, as well as influencing therapeutic resistance to achieve both carcinogenic and tumor suppressor effects [105,106]. For instance, the E3 ubiquitin ligase HDAC6 orchestrates the ubiquitination degradation of Chk1, thereby modulating the cell cycle and subsequently influencing radiosensitivity in non-small-cell lung cancer (NSCLC) [44]. In lung cancer tissues, Mxi1 exerts a negative regulatory influence on the oncogene Myc. However, the E3 ubiquitin ligase β-Trcp ubiquitinates and reduces Mxi1 protein levels, which leads to radioresistance in lung cancer [45]. Yang et al. made a further discovery, revealing that PRMT5 interacts with and methylates Mxi1. This event promotes the β-Trcp-mediated ubiquitination degradation of Mxi1, resulting in radioresistance [107]. Similarly, the E2/E3 hybrid ubiquitin–protein ligase ubiquitin-conjugating enzyme E2 O (UBE2O) mediates the ubiquitination degradation of Mxi1 [108]. The E3 ubiquitin ligase CHIP, however, impedes ionizing radiation-induced cellular senescence by mediating the ubiquitination degradation of p21. This process ultimately induces resistance to radiation in cancer cells [46]. However, a separate study found that CHIP could inhibit NSCLC stem cell properties and enhance radiosensitivity by inhibiting the PBK/ERK axis [47]. Additionally, another study demonstrated that CHIP could also regulate radiosensitivity by disrupting the interaction between Hsp90β and MAST1. This leads to the ubiquitination and downregulation of MAST1 stability, inhibiting the stemness of NSCLC stem cells [48]. Similarly, FOXN2 can modulate cell cycle redistribution to influence the sensitivity of lung cancer cells to radiation. FOXN2 depletion results in an increase in the number of S-phase cells. Further experiments revealed that the E3 ubiquitin ligase β-Trcp interacts with the RSK2 kinase, directly targeting the ubiquitin degradation of FOXN2. This further modulates cell cycle redistribution and cell proliferation, ultimately promoting lung cancer radioresistance [49]. PPDPF induces radioresistance in lung cancer cells by inhibiting apoptosis through the inhibition of BABAM2 degradation mediated by the E3 ligase MDM2 [50]. Cui et al. discovered that the deletion of FBXL14 promotes the expression of TWIST1 in NSCLC after radiation exposure, subsequently inducing the epithelial-to-mesenchymal transition (EMT) to promote radioresistance in cancer cells [109]. However, in lung adenocarcinoma cells (LUAD), the E3 ubiquitin ligase TRIM36 enhances radiosensitivity by promoting RAD51 ubiquitination and regulating DNA damage repair and apoptosis [51]. Another E3 ligase, FBXW7, inhibits NSCLC apoptosis and enhances radiosensitivity by targeting the SOX9/CDKN1A axis for ubiquitination [52]. Similarly, the E3 ligase UBR5 inhibits radiosensitization in NSCLC through the activation of the PI3K/AKT pathway [53]. Additionally, CDK20 competes with NRF2 for E3 ubiquitin ligase KEAP1 binding, enhances the transcriptional activity of NRF2, and participates in the oxidative stress response. This ultimately reduces ROS levels in the cells, leading to radioresistance in lung cancer cells [54]. The E3 enzyme FBXO22 mediates the ubiquitin degradation of PD-L1, increasing the sensitivity of NSCLC to IR and cisplatin [110]. Beyond common E3 enzymes, the ubiquitin-conjugating enzyme E2T (UBE2T), highly expressed in NSCLC tumor tissues, induces the ubiquitin degradation of FOXO1 and activates the downstream Wnt/β-catenin signaling pathway. This promotes NSCLC proliferation, the EMT, and radiation resistance [55].

In addition to ubiquitination, DUBs also play a crucial role in modulating the radiosensitivity of lung cancer tissues. For instance, USP9X can inhibit the ubiquitination degradation of its downstream target, KDM4C, through deubiquitination. This event regulates DNA damage repair via the TGF-β2/Smad/ATM signaling pathway, ultimately inhibiting cellular radiosensitivity [56]. Another study uncovered an alternative mechanism through which USP9X regulates the radiosensitivity of cancer cells in NSCLC. USP9X maintains the stabilization of the anti-apoptotic protein MCL1 which, in turn, inhibits apoptosis [57]. Similarly, the DUB USP39 stabilizes CHK2 (checkpoint kinase 2) via deubiquitination. CHK2, in turn, promotes the sensitivity of cancer tissues to radiotherapy by regulating apoptosis and cell cycle checkpoints after DNA damage. However, this study also demonstrated downregulated levels of USP39 and CHK2 in lung cancer cells, which may contribute to resistance to radiotherapy treatment in lung cancer tissues [58]. Xu et al. were the first to discover that the DUB UCHL3 was upregulated in NSCLC tissues and cells. UCHL3 maintains the stability of AhR proteins through deubiquitination, resulting in increased PD-L1 expression and enhanced radioresistance of NSCLC cells. The researchers further explored the upstream mechanism and confirmed that LINC00665 sponges miR-582-5p, thereby upregulating UCHL3. This raised the possibility of targeting the LINC00665/miR-582-5p/UCHL3/AhR axis to regulate radiosensitivity in NSCLC cells [59]. Liu et al. performed DUB UCHL3 knockdown to inhibit RAD51-mediated DNA damage repair, leading to the radiosensitization of NSCLC cells [60]. Another study found that the downregulation of the DUB USP14 led to an increase in NHEJ and a lack of HR, resulting in an imbalance in the DSB repair pathway and a failure to repair damaged DNA. This made NSCLC cells more sensitive to IR-mediated cell death [61].

### 2.4. Breast Cancer

Breast cancer is one of the most prevalent malignancies in women, and radiotherapy is a widely employed and effective clinical treatment, particularly for patients undergoing breast-conserving surgery [111,112]. An analysis of the Gene Expression Omnibus (GEO) databases GSE31863 and GSE101920 revealed the upregulation of the E3 ligase UBE3C in breast cancer tissue samples, and its elevation correlated with adverse radiological responses. Correlation experiments were conducted on molecules upstream and downstream of UBE3C; LINC00963 activates UBE3C transcription by facilitating the nuclear translocation of FOSB, and UBE3C catalyzes the ubiquitination degradation of the tumor suppressor TP73, thus enhancing the radioresistance of tumor cells [62]. Similarly, ubiquitin-conjugating enzyme E2 C (UBE2C) also plays a role in regulating the radiosensitivity of breast cancer cells, though the exact mechanism requires further exploration [113]. The E2 family member UBE2D3 not only impacts radiosensitivity in esophageal cancer [39] but also in breast cancer. UBE2D3 modulates telomerase activity and the cell cycle by reducing the expression of telomerase components hTERT and cyclin D1, leading to increased radiosensitivity [63]. β1-integrins have been shown to regulate breast cancer cell migration and mediate resistance to radiotherapy [114,115]. They were found to reduce the ubiquitylation of Rad51, a key factor in DNA damage repair, by modulating the level of the ubiquitin–protein ligase E3 RING1. This reduction in the ubiquitination degradation of Rad51 promotes DNA damage repair and contributes to radiotherapy resistance [64]. However, another study discovered that the DUB UCHL3 targets RAD51 for deubiquitination. Its role in this context primarily affects the function of RAD51 rather than the stability of the protein. This occurs through the promotion of RAD51 binding to BRCA2 and the facilitation of RAD51 aggregation at DSBs, leading to radiation resistance in cancer cells [67]. The E3 enzyme SKP2 fosters radiation tolerance in breast cancer by inhibiting apoptosis and promoting DNA damage response through PDCD4 ubiquitination degradation [65]. In triple-negative breast cancer (TNBC), the E3 ubiquitin ligase RNF126 mediates MRE11 ubiquitination. Instead of inducing degradation, RNF126 enhances its DNA exonuclease activity, activating the ATR-CHK1 cascade signaling for DNA damage repair. This confers resistance to radiotherapy in TNBC. Additionally, for the first time, it was found that IR induces the expression of RNF126 by activating the HER2-AKT-NF-κB pathway [66]. One study found that miR-205 was downregulated in radioresistant breast cancer cells. Further investigation revealed that miR-205 inhibits DNA damage repair by targeting ZEB1 and Ubc13, thereby allowing radiosensitization [116]. Another study demonstrated that ENO1 was associated with radioresistance in breast cancer cells through an analysis of data from the GEO database. Experimentally, it was shown that ENO1 enhances radioresistance in breast cancer by regulating mitochondrial homeostasis to reduce ROS production and inhibit apoptosis. Furthermore, LINC00663, an upstream regulator of ENO1, affects IR resistance by enhancing the E6AP-mediated DNA damage repair of ENO1 [117]. In TNBC, the expression of the E3 ubiquitin ligase TRIM32 is upregulated and promotes radiotherapy resistance. Mechanistically, TRIM32 could compete with TC45 for STAT3 binding, thereby inhibiting the TC45-mediated dephosphorylation of STAT3 and maintaining STAT3 activity, ultimately promoting TNBC radiotherapy resistance [118].

Additionally, deubiquitination plays a crucial role in regulating radiosensitivity. Recent studies have confirmed that CHK1 (checkpoint kinase 1) can promote the EMT in TNBC [119]. However, it has been demonstrated that the DUB USP7 catalyzes the deubiquitination of CHK1, facilitating DNA damage repair and leading to radioresistance. ZEB1 promotes interactions between USP7 and CHK1 [68]. Wang et al. proposed the involvement of the miR-200c/LINC02582/USP7/CHK1 signaling axis in regulating radiation resistance [120]. Following DNA damage, the DUB USP37 is phosphorylated by ATM. This promotes USP37 to respond to the DNA damage response (DDR) by maintaining the stability of BLM through the deubiquitination pathway, resulting in a decrease in IR sensitivity [69]. Similarly, the DUB OTUD6A is recruited to the site of DNA damage after dephosphorylation by PP2A. OTUD6A responds to DNA damage by blocking Top BP1 polyubiquitination, thus maintaining its stability. This subsequently promotes the phosphorylation of CHK1, regulating the cell cycle and leading to tumor cell resistance to chemoradiotherapy. Researchers also demonstrated that when OTUD6A is absent, mice become sensitive to IR [70].

In the context of the ubiquitination system, it is worth mentioning the ubiquitin-editing enzyme A20, which primarily regulates inflammation and immunity in an organism [121] and has also been found to be involved in tumor development [122]. However, it has been observed that A20 is upregulated in invasive breast cancer. When A20 is knocked down, there is an increase in NHEJ activity and a decrease in HR, disrupting the homeostasis of the DNA damage repair pathway. This promotes sensitivity to DNA damage and improves the sensitivity of cancer cells to radiotherapy. Mechanistically, A20 interacts with H2A by disrupting the E3 enzyme RNF168, thereby regulating DNA damage repair and maintaining genome stability [123].

### 2.5. Hepatocellular Carcinoma

Radiotherapy serves as a localized treatment option for hepatocellular carcinoma (HCC), one of the prevalent types of solid malignancies. The E2-conjugating enzyme UBE2T modulates cell cycle arrest by facilitating H2AX ubiquitination modification. This action activates CHK1, leading to the radioresistance of HCC cells [71]. Pro- and anti-apoptotic proteins coexist within tumor tissues [124]. Specifically, in P53-mutant HCC tissues, the E3 ubiquitin ligase CDC20 disrupts the Bax/Bcl-2 balance to avoid apoptosis and regulates cell cycle blockage, resulting in radiation resistance in HCC cells [125]. Ferroptosis is one of the main ways to kill HCC cells via radiation. COMMD10 disrupts Cu-Fe homeostasis in HCC cells, thus regulating the ubiquitination degradation of HIF1α. This inhibition of the HIF1α/CP loop enhances ferroptosis and radiosensitization [126]. Another study found that radiation sensitizes ferroptosis which, in turn, contributes to the radiation, “reverse” promoting the killing of HCC cells. This process involves the E3 ubiquitin ligase SOCS2, which mediates the polyubiquitination degradation of the downstream molecule SLC7A11, promoting ferroptosis and ultimately radiosensitizing HCC cells [72]. Previous research indicates that FoxA1 plays a crucial role in regulating the EMT [127]. Further investigation revealed that the E3 ubiquitin ligase RNF6 triggers the ubiquitination degradation of FoxA1, thus activating the EMT in HCC cells and causing radioresistance [73]. The long non-coding RNA NEAT1 is a source of radioresistance in HCC cells. Specifically, NEAT1v1 protects these cells from radiation-induced oxidative stress by boosting the mitochondrial localization of PINK1 and upregulating Parkin expression. This activates PINK1/Parkin pathway-mediated ubiquitin-dependent mitochondrial autophagy and protects HCC cells from radiation-induced oxidative stress, involving key factors such as GABARAP (a key factor in mitochondrial autophagy) and the antioxidant enzyme SOD2 [128]. In both HCC and colon cancer, PXR promotes MDM2 auto-ubiquitination, impairing MDM2-mediated ATF3 protein degradation. This enhances ATF3-mediated ATM activation in response to DNA damage, leading to radiation resistance [129]. Lastly, CPS1 is a key enzyme in the hepatic urea cycle, and its expression is downregulated in HCC cells. CPS1 silencing contributes to radiation resistance via c-Myc stability mediated by the ubiquitin–proteasome system [130].

### 2.6. Colorectal Cancer

Colorectal cancer is one of the gastrointestinal tumors with the highest incidence. Owing to atypical early symptoms, most patients are diagnosed at an advanced stage [131]. Radiotherapy plays a crucial role in treating patients with colorectal cancer, particularly those with advanced cases. Elevated levels of ubiquitinated proteins in human colorectal cancer SW620 cells follow C-ion irradiation. Treatment with proteasome inhibitors could enhance cell sensitivity to C-ion irradiation [132]. This suggests that ubiquitination and deubiquitination modifications potentially play significant roles in modulating the efficacy of radiotherapy in colorectal cancer. For instance, in rectal cancer, UBE2B can decrease cellular sensitivity to radiation by modulating DNA damage repair [133]. It has been shown that radiation in colorectal cancer prompts the E3 ligase FBW7 to target Mcl-1 for ubiquitination degradation. However, in the absence of the E3 ligase Skp2, this process can be facilitated, thereby enhancing the sensitivity of colorectal cancer (CRC) cells to radiotherapy. The exact mechanism underlying how Skp2 deficiency promotes interactions between FBW7 and Mcl-1 needs further exploration [134]. Another study revealed that the E3 ligase TRAF4 promotes Jun N-terminal kinase (JNK) ubiquitination, subsequently triggering the JNK/c-Jun signaling pathway. This leads to the activation of the transcription of the anti-apoptotic protein Bcl-xL, which drives radioresistance in CRC cells [74]. In addition to apoptosis, cellular autophagy plays a crucial role in colorectal cancer development [135]. ATG3 plays a pivotal role in the cellular autophagy pathway [136]. Further studies have revealed that in colorectal cancer, the long non-coding RNA (lncRNA) SP100-AS1 regulates cellular autophagy by modulating the level of ATG3 ubiquitylation. This stabilization of the ATG3 protein level attenuates the radiosensitivity of colorectal cancer. This experiment also suggested another mechanism by which SP100-AS1 affects autophagic activity—by acting as a sponge for miR-622 to directly stabilize ATG3, while miR-622 targets ATG3 mRNA [137]. Researchers initially examined pre-treatment specimens from patients with locally advanced rectal cancer. They found that patients with low levels of the E3 ubiquitin ligase RAD18 responded better to neoadjuvant chemoradiotherapy (nCRT). This suggests that RAD18 may potentially serve as a predictor of nCRT efficacy. Further investigation confirmed that the downregulation of RAD18 enhances cell radiosensitivity and 5-Fu sensitivity, promoting apoptosis by activating the caspase-9-caspase-3 pathway [75]. Chen et al. demonstrated that the activation of G3BP2 by RIOK1-mediated phosphorylation modulates the p53 signaling pathway by promoting the ubiquitination of p53 by MDM2. This culminates in colorectal cancer’s context of radioresistance [76]. In colorectal cancer cells with mitochondrial dysfunction, SIRT3 mediates mitochondrial autophagy through the PINK1/Parkin pathway. It also inhibits the expression of the ubiquitin ligase RING1b, further suppressing RING1b-mediated H2A ubiquitination and promoting DNA damage repair, resulting in radiation resistance in tumor cells [77]. In colorectal cancer, RBBP6 heightens radiation resistance. These researchers suggest that RBBP6 may regulate radiosensitivity, in part, by modulating the MDM2-mediated degradation of p53 ubiquitination [138].

### 2.7. Cervical Cancer

Cervical cancer, prevalent among middle-aged and elderly women, underscores the importance of radiotherapy in comprehensive treatment. Previous studies have shown that the ubiquitin system exhibits abnormal immune expression in HPV-positive cervical cancer tissues [139]. In this process, the E6-associated protein (E6AP) acts as an E3 ligase, mediating the degradation of p53 through the proteasome, thereby influencing the progression of cancerous tissues [140]. Further research revealed that upstream miR-375 can downregulate the ubiquitin ligase E3A (UBE3A), that is, E6AP, which subsequently affects the expression of the downstream factor p53, ultimately promoting the radiosensitization of cancer cells [141]. Moreover, in cervical cancer cells, the E3 ubiquitin ligase FBXO6 orchestrates the ubiquitination degradation of CD147. This promotes sensitivity to IR in radiotherapy-resistant cancer cells. However, heat shock protein 90 (HSP90) hinders this process, mediating radiotherapy resistance [78]. In addition to the ubiquitination system, the deubiquitination system also holds significance. It has been demonstrated that the DUB USP53 upregulates DDB2, facilitating DNA damage repair. Conversely, USP53 regulates the expression of the cell-cycle-associated protein CDK1, leading to radioresistance [142]. Additionally, the DUB OTUD5 leads to radiosensitization by decreasing the ubiquitination level of the signaling molecule Akt. This, in turn, affects Akt downstream molecules. However, further studies are needed to target downstream regulatory molecules [79]. Similarly, the DUB USP21 activates YAP1 by negatively regulating the ubiquitination of FOXM1. This inhibits Hippo signaling, thereby promoting radioresistance [80].

### 2.8. Head and Neck Squamous Cell Carcinoma (HNSCC) (Excluding NPC)

In head and neck squamous cell carcinoma, researchers found that the DUB BAP1 targets the substrate H2Aub for deubiquitination. This activity leads to radioresistance by promoting DNA damage repair [81]. Another research team specifically studied laryngeal cancer and discovered that the E3 ligase UBR5 inhibits radiosensitization by modulating the P38-MAPK signaling pathway [82]. In a separate study focused on laryngeal squamous cell carcinoma (LSCC), a significant increase in the expression of the DUB USP7 was observed in irradiated LSCC cells. This suggests that USP7 may influence the effectiveness of LSCC radiotherapy. As most patients with LSCC carry p53 mutations, which are important downstream genes of USP7, researchers examined the role of USP7 in radioresistance. They found that knocking down USP7 increased the radiosensitivity of p53-mutant LSCC cells but decreased the radiosensitivity of p53 wild-type cells. Further exploration is needed to understand how the USP7-p53 downstream pathway regulates radiosensitization [143]. In the case of oral squamous cell carcinoma, the E3 ligase TRAF4 activates Akt through the ubiquitination pathway, inhibits GSK3β activity and MCL-1 phosphorylation, and enhances the regulatory effects of the DUB JOSD1 on MCL-1. This series of actions ultimately increases the stability and expression of MCL-1, conferring cellular resistance to radiotherapy [83].

### 2.9. Central Nervous System (CNS) Tumors

Studies on central nervous system (CNS) tumors, specifically medulloblastoma, have shown that the E3 ubiquitin ligase RNF8 mediates PCNA ubiquitination. This action affects DNA damage repair and decreases the sensitivity of cancer cells to ionizing radiation, and it can regulate cell cycle and inhibit apoptosis [84]. In gliomas, two mechanisms have been identified. First, the E3 enzyme RAD18 confers radiation resistance to glioma cells by inhibiting apoptosis and regulating DNA damage repair [85]. Second, the E3 ligase HACE1 enhances the protein stability of NRF2 by competitively binding to NRF2 with another E3 ligase, KEAP1, and it also promotes the IRES-mediated translation of NRF2 mRNA together with the upregulation of NRF2. NRF2 reduces glioma cells’ sensitivity to radiation by decreasing cellular ROS levels. However, the above biological process is notably independent of HACE1’s E3 ligase activity [86]. Additionally, IRAK1 promotes radioresistance by inhibiting the E3 ubiquitin ligase HECTD3-mediated ubiquitination degradation of PRDX1. This stabilization of PRDX1 reduces cellular autophagy, contributing to radioresistance [87]. Similarly, linc-RA1 inhibits the interaction between H2Bub1 and the DUB USP44 to stabilize H2Bub1 levels, thereby inhibiting autophagy and contributing to glioma radioresistance [89]. A study focusing on glioma stem-cell-like cells found that G0S2 regulates lipid droplet turnover and inhibits the E3 enzyme RNF168-mediated ubiquitination of 53BP1 through the mTOR-S6K signaling pathway. This promotes 53BP1’s response to ionizing radiation, enhancing DNA damage repair and glioma radioprotection [88]. In highly malignant glioblastomas, the E3 ubiquitin ligase RNF138 has been shown to mediate ribosomal protein S3 (rpS3) ubiquitination, thereby inhibiting rpS3/DDIT3-mediated apoptotic signaling when stimulated by radiation and inducing radioresistance in glioblastoma (GBM) cells [90]. A comparison of various glioblastoma cell lines revealed that the DUB USP9x affects cell survival by regulating Mcl-1 levels in some cell lines. However, in others, USP9x’s role in radiosensitization was found to be independent of Mcl-1 levels. Researchers thus concluded that USP9x can regulate cellular radiosensitization through both Mcl-1-dependent and Mcl-1-independent mechanisms [144].

## 3. Discussion

Radiotherapy stands as a cornerstone in the personalized treatment of patients with cancer. However, inherent or acquired resistance to radiation is a major cause of the low efficacy of radiotherapy and significantly limits its effectiveness. Extensive research has shown that enzymes orchestrating ubiquitination and deubiquitination processes play pivotal roles in governing tumor behaviors, such as proliferation, migration, invasion, and therapeutic resistance. Ubiquitination, a widespread protein modification, hinges on three distinct enzymes for the covalent attachment of ubiquitin to substrate proteins, a process counteracted by DUBs.

This review delves into the advancements surrounding enzymes within the ubiquitination/deubiquitination system in modulating radiosensitization. This understanding paves the way for innovative strategies in radiosensitization, showing immense clinical potential. A prime example is the clinical application of the proteasome inhibitor bortezomib in treating multiple myeloma [145], underscoring the feasibility of targeting the ubiquitin–proteasome pathway in cancer therapy. Nevertheless, there remain challenges to be addressed through comprehensive future investigations. Firstly, ubiquitin catalytic enzymes, particularly E3 enzymes, exhibit vast diversity and are extensively involved in diverse biological processes. Mutations, inhibition, or overexpression in these enzymes can have far-reaching impacts on downstream biological processes, potentially leading to disease. Thus, when proposing treatment targeting a pivotal enzyme, its involvement in normal activities must be carefully considered. Secondly, while proteasome inhibitors in the ubiquitination system have gained traction in basic research for treating malignant tumors, especially hematological malignancies, their clinical application in solid tumors remains relatively limited [146]. Further in-depth clinical studies are imperative to validate their broader biological functions for effective translation into clinical practice. Moreover, studies examining whether these drugs influence radiosensitivity and whether they synergize with radiotherapy are scarce and in the early stages. For instance, the proteasome inhibitor MG132 has been shown to enhance the radiosensitivity of lung cancer cells [147], but this warrants verification through clinical experiments. Additionally, radiotherapy inevitably inflicts irreversible radiation damage on surrounding normal tissues. Consequently, changes in the activity of enzymes mediating ubiquitination/deubiquitination after radiotherapy raise questions about potential links to radiation damage. Can these enzymes be targeted to minimize radiation-induced harm? Lastly, given the intricate network of mechanisms involving ubiquitination/deubiquitination in regulating radiosensitivity, it is evident that for the same tumor, such as lung cancer, multiple enzymes collectively regulate radiosensitivity. However, the same enzyme may function through a variety of substrate proteins; even for different tumors, the same enzyme can regulate radiosensitivity (Figure 2). The influences of the factors are intricate and complex, suggesting that if we want to intervene in radiosensitivity, the effect of targeting a certain enzyme individually may yield unsatisfactory results. Therefore, it is reasonable to posit that identifying a more upstream or downstream co-factor may yield superior outcomes. In the future, we also hope to explore a broader spectrum of key enzymes to intervene in radiosensitivity, benefiting a larger number of patients.

While significant strides have been made in ubiquitination/deubiquitination research, the clinical translation of therapeutics targeting ubiquitination is still a journey fraught with challenges. It necessitates concerted efforts from researchers. In the future, delving into the undiscovered mechanisms through which various ubiquitinating enzymes and DUBs influence radiosensitivity remains paramount. This involves further exploration of their upstream/downstream target molecules. Identifying meaningful ubiquitinating enzymes and DUBs as prognostic indicators for radiotherapy or as targets for antitumor drugs holds promise for future advancements in cancer treatment. 

## 4. Conclusions

Radiation resistance has always been a detrimental factor in the efficacy of radiotherapy. Increasing the radiation dose may improve local control in patients, but this approach is often abandoned due to increased damage to surrounding normal tissues. Promoting the radiosensitivity of tissues may be a favorable approach for local control. Here, we emphasize that the enzymes involved in ubiquitination/deubiquitination are important factors in regulating radiosensitivity. On the one hand, most studies indicate that the expression of certain catalytic enzymes can mediate the development of radioresistance in tumors. On the other hand, some enzymes increase radiosensitivity through various pathways. This suggests that the ubiquitination/deubiquitination system has the potential to become a target for enhancing the effectiveness of radiation therapy and a biomarker for predicting the efficacy of combination therapy. We believe that with the continuous advancement of technology, drugs targeting the ubiquitination/deubiquitination system can be reasonably applied in clinical practice, benefiting more cancer patients.

## Figures and Tables

**Figure 1 biomedicines-11-03240-f001:**
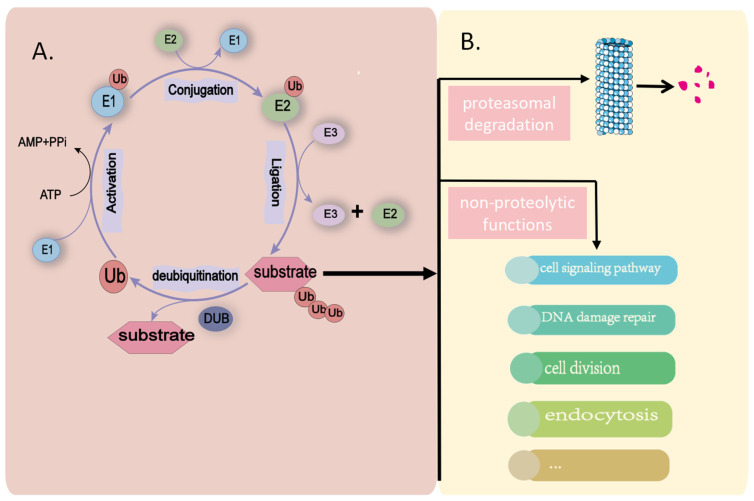
The biological processes and functions of the ubiquitination and deubiquitination sys-tems. (**A**) Ubiquitin is first activated by an E1 enzyme, after which the E1 enzyme passes the activated ubiquitin to an E2 enzyme, and the activated ubiquitin is then ligated or tagged to a substrate catalyzed by an E3 enzyme. Deubiquitinase removes ubiquitin from the substrate, thereby reversing the ubiquitination modification of the substrate. (**B**) Ubiquitin modification primarily mediates protein degradation and regulates protein levels. Additionally, ubiquitin modification also possesses non-proteolytic functions, including involvement in cell signaling pathways, DNA damage repair, cell division, and endocytosis. (E1: ubiquitin-activating enzyme; E2: ubiquitin-conjugating enzyme; E3: ubiquitin ligase; Ub: ubiquitin; ATP: adenosine triphosphate; AMP: adenosine monophosphate; PPi: pyrophosphoric acid; DUB: deubiquitinase).

**Figure 2 biomedicines-11-03240-f002:**
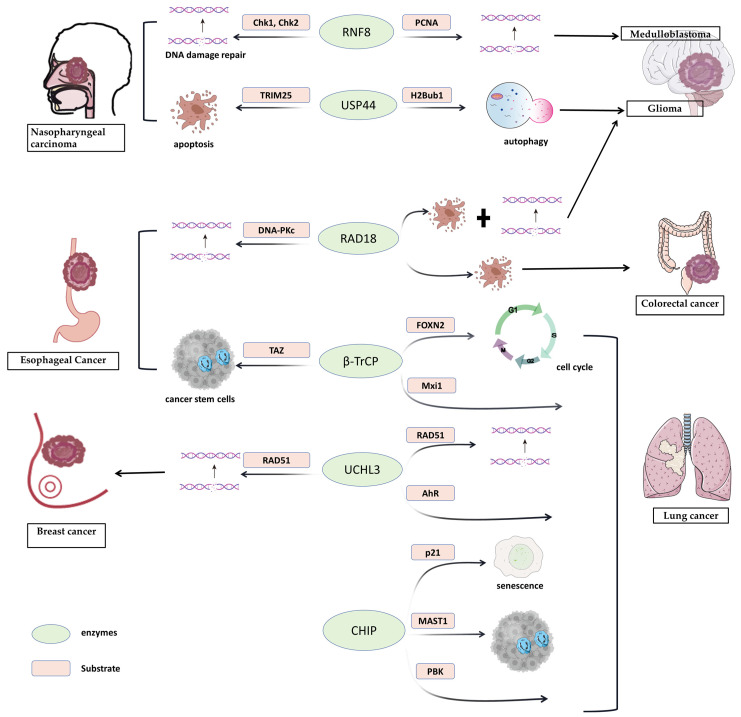
The same enzymes and/or mechanisms regulating radiosensitivity present in two or more different tumor types. The same enzymes interact with different substrate proteins in different tumor types, thereby regulating tumor radiosensitivity through various mechanisms, such as DNA damage repair, apoptosis, and autophagy. (RNF8: Ring finger protein 8; Chk1 and Chk2: cell-cycle regulation kinases; PCNA: Proliferating cell nuclear antigen; USP44: ubiquitin-specific protease 44; TRIM25: tripartite motif-containing (TRIM) protein; H2Bub1: H2B K120 monoubiquitination; DNA-PKc: DNA-dependent protein kinase complex; β-Trcp: β-transducin repeat-containing protein; Mxi1: MAX interactor 1;AhR: aryl hydrocarbon receptor; CHIP: Carboxyl terminus of Hsc70-interacting protein; PBK: PDZ-binding kinase; MAST1: microtubule-associated serine/threonine kinase 1).

**Table 1 biomedicines-11-03240-t001:** The role of key enzymes in ubiquitination and deubiquitination in regulating the radiosensitivity of solid tumors.

Cancer Types	Enzyme Types	Enzyme	Substrate	Mechanism	References
Nasopharyngeal carcinoma	E3	TRIM21	GMPS	With the assistance of SERPINB5, TRIM21 mediates the ubiquitination and degradation of GMPS, leading to the suppression of TP53 expression and subsequently inhibiting cell apoptosis.	[33]
	E3	TRIM21	VDAC2	TRIM21 mediates the ubiquitination and degradation of VDAC2, effectively inhibiting the release of mitochondrial DNA (mtDNA) and consequently impairing the radiation-induced anti-tumor immune response–type-I interferon response.	[34]
	E3	RNF8	Chk1, Chk2	Multiple factors such as Chk1 and Chk2 are recruited and ubiquitinated by RNF8 to regulate their activity and stability, leading to DNA damage repair and resulting in radiotherapy resistance.	[35]
	E3	FBXW7	mTOR	FBP1 inhibits the autoubiquitination of FBXW7, leading to the promotion of downstream mTOR ubiquitination by FBXW7, which in turn inhibits mTOR levels and suppresses glycolysis, ultimately enhancing radiosensitivity	[36]
	deubiquitinase	USP44	TRIM25	USP44 targets the E3 ubiquitin ligase TRIM25 for deubiquitination, leading to the degradation of downstream Ku80 and inhibiting DNA damage repair involving Ku80. Additionally, USP44 regulates the cell cycle and induces apoptosis, ultimately enhancing radiation sensitivity.	[37]
	deubiquitinase	OTUD4	GSDME	OTUD4 stabilizes GSDME via deubiquitination, thereby promoting GSDME-mediated pyroptosis to enhance radiation sensitivity.	[38]
Esophageal Cancer	E2	UBE2D3	hTERT	When UBE2D3 is knocked down, it not only enhances the expression and activity of telomerase enzyme hTERT to promote telomere stability but also affects the cell cycle and DNA repair capacity, thereby inducing radiation resistance.	[39]
	E3	SOCS6	c-Kit	SOCS6 catalyzes the ubiquitination and degradation of c-Kit, affecting tumor cells’ stemness and thereby sensitizing them to radiotherapy.	[40]
	E3	RAD18	DNA-PKc	RAD18 promotes NHEJ-mediated repair of DSBs by upregulating the phosphorylation levels of DNA-PKc, resulting in resistance to radiation therapy.	[41]
	E3	PELI1	NIK	By inhibiting the activation of the NIK/NF-κB signaling pathway through ubiquitination, PELI1 promotes cancer cell apoptosis and ultimately enhances radiosensitivity.	[42]
	E3	β-TrCP	TAZ	TRIB3 induces radioresistance by promoting CSC properties through inhibiting β-TrCP-mediated TAZ ubiquitination and degradation.	[43]
Lung cancer	E3	HDAC6	Chk1	HDAC6 mediates the ubiquitination and degradation of Chk1, regulating radiation sensitivity by influencing the cell cycle.	[44]
	E3	β-Trcp	Mxi1	With the assistance of S6K1, β-Trcp can ubiquitinate and downregulate Mxi1 levels, affecting Mxi1’s negative regulation of the oncogene Myc, mediating radiation resistance in lung cancer.	[45]
	E3	CHIP	p21	CHIP mediates the ubiquitination and degradation of p21, inhibiting cellular senescence induced by ionizing radiation, thereby inducing radiation resistance.	[46]
	E3	CHIP	PBK	CHIP suppressed stem cell properties and the radioresistance of NSCLC cells by inhibiting the PBK/ERK axis.	[47]
	E3	CHIP	MAST1	CHIP disrupts the interaction between Hsp90β and MAST1 and ubiquitinates and downregulates MAST1 protein stability to inhibit the stemness of stem cells.	[48]
	E3	β-Trcp	FOXN2	β-Trcp interacts with RSK2 kinase, targeting FOXN2 for ubiquitination and degradation, thereby promoting radiation resistance by regulating cell cycle and cell proliferation.	[49]
	E3	MDM2	BABAM2	PPDPF inhibits cell apoptosis and induces resistance to radiation in lung cancer cells by suppressing the MDM2-mediated degradation of BABAM2.	[50]
	E3	TRIM36	RAD51	TRIM36 promotes the ubiquitination of RAD51, enhancing radiation sensitivity by regulating DNA repair and cell apoptosis.	[51]
	E3	FBXW7	SOX9	FBXW7 enhances radiation sensitivity by targeting the SOX9/CDKN1A axis through ubiquitination to inhibit cell apoptosis.	[52]
	E3	UBR5	—	UBR5 suppresses the sensitivity of cancer tissue to radiation by activating the PI3K/AKT pathway.	[53]
	E3	KEAP1	NRF2	CDK20 competitively binds the E3 ubiquitin ligase KEAP1 to NRF2, enhances the transcriptional activity of NRF2, and lowers the cellular reactive oxygen species level.	[54]
	E2	UBE2T	FOXO1	UBE2T induces ubiquitination and degradation of FOXO1, activating the downstream Wnt/β-catenin signaling pathway and promoting proliferation, EMT, and radiation resistance in NSCLC.	[55]
	Deubiquitinase	USP9X	KDM4C	USP9X regulates DNA damage repair by deubiquitinating KDM4C, thereby inhibiting cell sensitivity to radiation.	[56]
	Deubiquitinase	USP9X	MCL1	USP9X inhibits cell apoptosis by maintaining the stability of the anti-apoptotic protein MCL1.	[57]
	Deubiquitinase	USP39	CHK2	USP39 stabilizes CHK2 via deubiquitination, regulating cell apoptosis and the cell cycle after DNA damage, promoting sensitivity to radiotherapy and chemotherapy.	[58]
	Deubiquitinase	UCHL3	AhR	UCHL3 deubiquitinates and maintains the stability of AhR protein, thereby increasing PD-L1 expression and enhancing radioresistance.	[59]
	Deubiquitinase	UCHL3	RAD51	Knockdown of UCHL3 can inhibit RAD51-mediated DNA damage repair, leading to increased sensitivity of cancer cells to radiation.	[60]
	Deubiquitinase	USP14	—	Downregulation of USP14 leads to imbalances in DSB repair pathways (NHEJ and HR), resulting in ineffective repair of damaged DNA and making cancer cells more sensitive to cell death mediated by IR.	[61]
Breast cancer	E3	UBE3C	TP73	LINC00963 induces nuclear translocation of FOSB and the consequent transcription activation of UBE3C, which enhances radioresistance by inducing ubiquitination-dependent protein degradation of TP73.	[62]
	E2	UBE2D3	hTERT	UBE2D3 reduces the expression levels of hTERT and cyclin D1 to regulate telomerase activity and the cell cycle, thereby increasing radiosensitivity.	[63]
	E3	RING1	Rad51	β1-integrin regulates the protein level of RING1, reducing the ubiquitination and degradation of Rad51, thereby promoting DNA damage repair and leading to radiotherapy resistance.	[64]
	E3	SKP2	PDCD4	SKP2 promotes radiation tolerance by facilitating the ubiquitination and degradation of PDCD4, inhibiting cell apoptosis, and promoting DNA damage response.	[65]
	E3	RNF126	MRE11	RNF126 mediates the ubiquitination of MRE11, promoting its DNA exonuclease activity to activate the ATR-CHK1 signaling pathway for repairing damaged DNA, conferring resistance to radiotherapy in triple-negative breast cancer.	[66]
	Deubiquitinase	UCHL3	RAD51	UCHL3 targets RAD51 for deubiquitination, promoting the binding of RAD51 with BRCA2 and facilitating the aggregation of RAD51 at DNA double-strand breaks (DSBs), ultimately leading to radiation tolerance.	[67]
	Deubiquitinase	USP7	CHK1	USP7 catalyzes the deubiquitination of CHK1, promoting DNA damage repair and leading to radiation resistance.	[68]
	Deubiquitinase	USP37	BLM	During DNA damage, USP37 is phosphorylated by ATM, which in turn promotes the deubiquitination of BLM by USP37 to maintain the stability of BLM and respond to DNA damage response, resulting in decreased sensitivity to IR.	[69]
	Deubiquitinase	OTUD6A	TopBP1	OTUD6A responds to DNA damage by blocking the ubiquitination of TopBP1, subsequently promoting CHK1 phosphorylation and regulating the cell cycle, leading to resistance to radiotherapy and chemotherapy.	[70]
Hepatocellular carcinoma	E2	UBE2T	H2AX	UBE2T mediates the ubiquitination modification of H2AX, further activating CHK1, thereby regulating the cell cycle and leading to radiation resistance.	[71]
	E3	SOCS2	SLC7A11	SOCS2 mediates the ubiquitination and degradation of SLC7A11, thereby promoting ferroptosis and ultimately leading to radiation sensitivity.	[72]
	E3	RNF6	FoxA1	RNF6 can induce ubiquitination and degradation of FoxA1 to regulate EMT activation, leading to radiotherapy resistance.	[73]
Colorectal cancer	E3	TRAF4	JNK	TRAF4 promotes JNK ubiquitination, thereby triggering the JNK/c-Jun signaling pathway. c-Jun promotes the transcription of the anti-apoptotic protein Bcl-xL, driving radiation resistance.	[74]
	E3	RAD18	—	Downregulating RAD18 promotes cell apoptosis and enhances radiosensitivity by activating the caspase-9-caspase-3 pathway.	[75]
	E3	MDM2	p53	After being phosphorylated by RIOK1, G3BP2 is activated and promotes the MDM2-mediated ubiquitination of p53, leading to radioresistance through regulation of the p53 signaling pathway.	[76]
	E3	RING1b	H2A	SIRT3 mediates mitophagy through the PINK1/Parkin pathway, which subsequently suppresses the expression of RING1b. This suppression of RING1b inhibits H2A ubiquitination and enhances DNA damage repair, leading to increased resistance to radiation.	[77]
Cervical cancer	E3	FBXO6	CD147	FBXO6 mediates the ubiquitination and degradation of CD147 to promote cancer cell sensitivity to ionizing radiation (IR).	[78]
	Deubiquitinase	OTUD5	Akt	OTUD5 decreases the ubiquitination level of Akt and affects the downstream molecules of Akt, leading to radiosensitization.	[79]
	Deubiquitinase	USP21	FOXM1	USP21 activates YAP1 by deubiquitinating FOXM1, thereby inhibiting the Hippo signaling pathway to promote radioresistance.	[80]
Head and neck squamous cell cancer	Deubiquitinase	BAP1	H2Aub	BAP1 mediates the deubiquitination of H2Aub, thereby promoting DNA repair and leading to radioresistance.	[81]
	E3	UBR5	—	UBR5 inhibits radiosensitivity by regulating the P38-MAPK signaling pathway.	[82]
	E3	TRAF4	Akt	TRAF4 activates the Akt signaling pathway through the ubiquitination pathway and promotes the interaction between JOSD1 and MCL-1, collectively enhancing the stability of MCL-1 and conferring radiotherapy resistance.	[83]
Medulloblastoma	E3	RNF8	PCNA	RNF8 mediates PCNA ubiquitination, affecting DNA damage repair and reducing the sensitivity of cancer cells to ionizing radiation. It also regulates the cell cycle and inhibits apoptosis.	[84]
Glioma	E3	RAD18	—	RAD18 mediates radiation resistance by inhibiting cell apoptosis and regulating DNA damage repair.	[85]
	E3	HACE1	NRF2	HACE1 not only competes with the E3 ligase KEAP1 but also promotes the mRNA translation of NRF2, collectively upregulating the levels of NRF2. NRF2, through reducing cellular ROS levels, decreases the response of cells to radiation.	[86]
	E3	HECTD3	PRDX1	IRAK1 inhibits the HECTD3-mediated ubiquitination and degradation of PRDX1, thereby stabilizing PRDX1, which in turn promotes radioresistance by reducing cellular autophagy.	[87]
	E3	RNF168	53BP1	G0S2 regulates lipid droplet turnover, thereby activating the mTOR-S6K signaling pathway, attenuating the RNF168-mediated ubiquitination of 53BP1 and promoting the response of 53BP1 protein to ionizing radiation, resulting in radioresistance.	[88]
	Deubiquitinase	USP44	H2Bub1	linc-RA1 inhibits the interaction between H2Bub1 and USP44 to stabilize the level of H2Bub1, thereby inhibiting autophagy and contributing to radioresistance.	[89]
	E3	RNF138	rpS3	RNF138 inhibits the cell apoptosis signaling mediated by rpS3/DDIT3, thereby inducing radioresistance.	[90]

## Data Availability

The data supporting the conclusions of this review are included in the article.

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
