# Peer review of "The Mechanism of Ubiquitination or Deubiquitination Modifications in Regulating Solid Tumor Radiosensitivity"

_biomedicines, 2023, doi:10.3390/biomedicines11123240_

Round 1
Reviewer 1 Report
Comments and Suggestions for Authors
This paper discussed the use of radiotherapy to treat tumors and focused on its limitations to various drug resistance factors. For this, it has highlighted the dual role of the ubiquitination/deubiquitination system, a reversible molecular modification pathway, in influencing cancer cell survival.
The content is impressive, but there is one minor concern. Although wnt/beta-catenin signaling was discussed in the content, the mode of action of RNF146, a type of E3 ubiquitin ligase, was not included. It is well established that the role of RNF146 through ubiquitination of the LKB1-AMPK pathway is related to radiotherapy resistance (eg. esophageal cancer). Therefore, this reviewer suggests supplementing relevant content in appropriate sections or tables.
Comments on the Quality of English LanguageMinor language polishing is required.
Author Response
We sincerely appreciate the valuable comments. We have checked the literature carefully and added relevant references on RNF146 and LKB1-AMPK pathway into the “2.2. Esophageal Cancer” part in the revised manuscript.
Reviewer 2 Report
Comments and Suggestions for Authors
General comments to the paper entitled: The Mechanism of Ubiquitination or Deubiqutination Modification in Regulating Solid Tumor Radiosensitivity
The ubiquitination/deubiquitination is a reversible molecular modification pathway that influences tumor behavior (cancer progression, proliferation, migration, invasion, and therapeutic resistance). This mechanism modulates the response to radiotherapy. The paper gives a comprehensive survey of the mechanisms of the ubiquitination/deubiquitination system's role in radiotherapy resistance in cancer tissue.
The authors systemically collected all the related papers discussing the ubiquitination/deubiquitination pathways, including the enzymes, genes, and signal systems involved in the mechanism in nine different cancer types. The paper clearly shows the complexity of the mechanisms in Table 1. which looks different in the nine different cancer lines but has some similarities.
As it is impossible to make some general conclusion, I suggest adding one more Table highlighting the same or similar enzymes and/or mechanisms present in two or more different cancer types. This may indicate the critical mechanism to focus if a reader plans to be involved in this research field.
Author Response
Thank you for your nice suggestion. Based on your suggestion, we have included another Figure 2 in the article, highlighting the enzymes, substrates, and related mechanisms that can simultaneously regulate radiotherapy sensitivity in multiple cancer types. In the discussion section of this article, we also mentioned that it is due to the existence of these intricate and interconnected factors that searching for molecules upstream or downstream may lead to better outcomes. In the future, we also hope to explore a broader spectrum of key enzymes to intervene in radiosensitivity, benefiting a larger number of patients.
Reviewer 3 Report
Comments and Suggestions for Authors
This manuscript is an enumeration of data from an extensive and relevant literature. As such it is useful. However, it lacks a critical approach to the experimental or clinical observations underpinning these data.
The follow comments might be useful for revision.
One. All abbreviations need to be explained in table and text.The list of abbreviations at the end of the manuscript (lines 526 to 533) is far from complete.
Two. Figure 1: what is the difference between the upper (degradation) and the lower arrow (DNA damage amongst others)? Figures should be readable without the text; so, explain all symbols and abbreviations in the legend.
Three. The Terminology should be uniform. Examples: figure 1 and text Lines 76 to 78 for “biological processes”; “human cells” to illustrate ubiquitination on line 50 and “the organism” for deubiquitination on line 86; figure 1 and figure 2. And there are many other examples.
Four. Redundancy: Figure 2 duplicates figure 1. The content of Table 1 is almost literally repeated in the text. Wouldn’t a graphical presentation of the pathways involved be more readable than the Table?
Author Response
Thank you for your nice suggestion.
One. We are sorry about our mistake. We have made updates to ensure the standardized use of abbreviations.
Two. According to your suggestion, we have reorganized and annotated the legends in the manuscript.
Three. Thank you so much for your careful check. We have standardized the terminology used in the manuscript.
Four. Thanks for your suggestion. There was indeed some overlap between Figure 1 and Figure 2, and we have removed Figure 2. We have created a new Figure 2 to visually represent some of the content from Table 1. Although Table 1 overlaps with the main text, some readers may prefer to refer to tables while others prefer the main text. Therefore, we have decided to keep Table 1 as it is.
Round 2
Reviewer 3 Report
Comments and Suggestions for Authors
Nihil